# Role of DNA Methylation Profiles as Potential Biomarkers and Novel Therapeutic Targets in Head and Neck Cancer

**DOI:** 10.3390/cancers15194685

**Published:** 2023-09-22

**Authors:** Kyunghee Burkitt

**Affiliations:** Head and Neck Medical Oncology, University Hospitals Cleveland Medical Center, Case Western Reserve University School of Medicine, Cleveland, OH 44106, USA; kyunghee.burkitt@uhhospitals.org

**Keywords:** head and neck squamous cell carcinoma (HNSCC), DNA methylation, HPV-positive HNSCC, HPV-negative HNSCC, oropharyngeal squamous cell carcinoma (OPSCC), DNA methylation/demethylation, DNA methyltransferase (DNMT) inhibitors, potential biomarkers: tumorigenesis/diagnostic/prognostic, tumor immune microenvironment (TIME), immunotherapy, programmed-death-1 (PD-1) inhibitor, cytotoxic T-lymphocyte associated protein 4 (CTLA-4) inhibitor, overall survival (OS), recurrence-free survival (RFS)

## Abstract

**Simple Summary:**

Recent studies have shown that DNA methylation alters gene expression, which drives tumorigenesis in cancer, and also, increasing evidence has shown that DNA methylation can regulate the tumor immune microenvironment (TIME) of cancer, which can affect the response to immunotherapy. For these reasons, DNA methylation has been recently recognized as a potential therapeutic target in various cancers, including head and neck cancer. Epigenetic therapy, specifically the role of DNA methyltransferase (DNMT) inhibitors in head and neck cancer treatment, is still in a primitive stage; therefore, a review of the published evidence of known DNA methylation profiles in head and neck cancer is important to access the potential value of DNA methylation as a diagnostic, prognostic, and predictive biomarker. Ultimately, I envision the rational design of clinical trials that will target a select group of head and neck patients with DNA methylation alterations that can be targeted in combination with chemotherapy and/or radiotherapy or immunotherapy.

**Abstract:**

Head and neck squamous cell carcinoma (HNSCC) is the sixth most common cancer worldwide and is associated with high mortality. The main reasons for treatment failure are a low rate of early diagnosis, high relapse rates, and distant metastasis with poor outcomes. These are largely due to a lack of diagnostic, prognostic, and predictive biomarkers in HNSCC. DNA methylation has been demonstrated to play an important role in the pathogenesis of HNSCC, and recent studies have also valued DNA methylation as a potential biomarker in HNSCC. This review summarizes the current knowledge on DNA methylation profiles in HPV-positive and HPV-negative HNSCC and how these may contribute to the pathogenesis of HNSCC. It also summarizes the potential value of DNA methylation as a biomarker in the diagnosis, prognosis, and prediction of the response to therapy. With the recent immunotherapy era in head and neck treatment, new strategies to improve immune responses by modulating TIMEs have been intensely investigated in early-phase trials. Therefore, this study additionally summarizes the role of DNA methylation in the regulation of TIMEs and potential predictive immunotherapy response biomarkers. Finally, this study reviews ongoing clinical trials using DNA methylation inhibitors in HNSCC.

## 1. Introduction

Head and neck squamous cell carcinoma (HNSCC) is the sixth most common cancer worldwide with approximately 60,000 patients diagnosed annually in the United States [1]. While patients with human papilloma virus (HPV)-positive HNSCC tend to have an excellent prognosis [2], HPV-negative patients have an approximately 50% recurrence rate despite treatment with standard cisplatin-based chemoradiotherapy and/or surgery. Furthermore, patients with recurrent/metastatic (R/M) HNSCC treated with platinum-based chemotherapy regimens have a dismal prognosis with a median overall survival of approximately 10 months [3]. Recently, pembrolizumab, which blocks the programmed-death-1 (PD-1)/programmed death-ligand 1 (PD-L1) axis, was approved as a first-line therapy for patients with R/M HNSCC either as monotherapy for PD-L1 positive patients (combined positive score, CPS ≥ 1) or in combination with platinum and 5-fluorouracil [4]. Nivolumab is another anti-PD1 inhibitor approved for R/M HNSCC patients who are refractory to platinum-based chemotherapy within 6 months. However, the response rates to pembrolizumab and nivolumab monotherapy are as low as 19% and 13.3%, respectively, underscoring the presence of immune evasion mechanisms [5]. A lack of diagnostic, prognostic, and predictive biomarkers in HNSCC is also the main reason why clinicians have difficulty in diagnosing or detecting recurrent/metastatic disease at an early stage.

Considering the limits of the current standard of care treatment as mentioned above, a deeper understanding of the underlying biological differences between the tumorigenesis of HPV-positive- and HPV-negative HNSCC and also understating the immune evasion mechanism will provide rationales to successfully design clinical trials with the goal to ultimately improve the current standard therapy in HNSCC. In the past, multiple studies suggested that DNA methylation alters the gene expression that drives tumorigenesis and tumor progression in cancers, including head and neck cancer [6,7,8]. More recently, it has been shown that DNA methylation can regulate the tumor immune microenvironment (TIME) of cancer, which can affect the response to immunotherapy [9]. Specifically, a recent study showed that de novo DNA methylation renders terminal T cell exhaustion, and subsequent exhausted T cells are refractory to PD-1 blockade-mediated rejuvenation [10]. Pretreatment with DNA methyltransferases (DNMT) inhibitor prior to PD-L1 blockade showed an enhancement of T cell responses and tumor control during PD-1 inhabitation in mice [10,11]. 

Currently, HPV status is the only available diagnostic and prognostic biomarker for HPV-positive HNSCC. There is still a significant unmet need for diagnostic, prognostic, and predictive biomarkers in HNSCC, especially HPV-negative HNSCC, in order to manage the disease more effectively. Therefore, I summarize the current knowledge on DNA methylation profiles in both HPV-positive and HPV-negative HNSCC in order to highlight specific DNA methylation profiles that can have the potential to constitute clinically valuable biomarkers in the management of head and neck cancer patients.

In addition, the present review discusses the most updated understanding of the role of DNA methylation in the tumor immune microenvironment (TIME) of HNSCC and its role as a potential biomarker for the prediction of the response to immunotherapy, such as PD1 inhibitors or CTLA-4 (cytotoxic T-lymphocyte associated protein 4) immune checkpoint inhibitors. Lastly, ongoing clinical trials evaluating DNMT inhibitors (azacitidne, decitabine) in the treatment of head and neck cancer patients are also reviewed.

From a clinical perspective, a present review of the role of DNA methylation as a potential biomarker will help clinicians consider valuable DNA methylation biomarkers for improving current standard care treatments. In addition, applying DNA methylation as a novel therapeutic target either with chemotherapy or immunotherapy in clinical trials will ultimately improve the overall survival of head and neck cancer patients.

## 2. DNA Methylation Profiles in HNSCC

### 2.1. DNA Methylation Profiles in HPV-Positive HNSCC

HPV-positive HNSCC is known to have a higher frequency of aberrant DNA methylation compared to HPV-negative HNSCC. The mechanism underlying increased DNA methylation in HPV-positive HNSCC has been well demonstrated in the literature. HPV is known to encode two oncoproteins, E6 and E7, which are involved in HPV-related malignancies, such as HNSCC and cervical cancer [12]. The HPV E6 oncoprotein directly targets p53 and induces p53 degradation by ubiquitination. This results in increased DNMT1 expression, which is normally suppressed by p53. HPV E6 also forms a complex with DNMT 1, which increases DNMT1 expression [13]. In addition, *c-Myc* (MYC) is also reported to recruit DNMT3A and the MYC-associated genetic network, which is known to be activated in HPV-associated HNSCC [13]. 

Different studies have shown that various genes are frequently hypermethylated in HPV-positive HNSCC (Table 1). These hypermethylated genes are involved in multiple different biological pathways, such as cell cycle regulation and apoptosis (*CCNA1*, *RASSF1*, *CDKN2A*), cellular adhesion (*CDH 1*, *8*, *11*, *13*, *15*, *18*, *19*, *23*, *CADM1*, *ITGA4*), cellular migration (*TIMP3*, *ELMO1*, *CTNNA2*), differentiation (*RXRG*, *GATA4*), and G protein-coupled receptor (GPCR) signaling (*HCRTR2*, *GALR1*, *HCRTR2*, *TACR1*, *NTSR1*). 

More recently, a study by Hinic et al. [14] using TCGA transcriptome and DNA methylome analysis in HPV-positive HNSCC patients identified 1854 differentially expressed genes, which were specifically compared to HPV-negative HNSCC patients [14]. Among these genes, *SYCP*2 and TAF7L were two of the top hypomethylated and upregulated deregulated genes in HPV-positive HNSCC. This finding draws special attention because a significant increase in *SYCP2* gene expression was observed in OPSCC compared to normal and premalignant tissues, and its role in tumorigenesis was suggested in another study [15]. *TAF7L* was previously suggested for its role in the tumorigenesis of breast cancer [16]. Further discussion on the role of these hypomethylated genes as potential biomarkers in tumorigenesis will be followed in Section 3.1.

### 2.2. DNA Methylation Profiles in HPV-Negative HNSCC

Compared to the DNA methylation profiles of HPV-positive HNSCC, fewer studies have been conducted to investigate the DNA methylation profiles of HPV-negative HNSCC. While p16-IHC has been used as a surrogate marker for HPV status and research is ongoing to find novel biomarkers for HNSCC to predict outcomes of treatment, biomarkers are more urgently needed for HPV-negative HNSCC patients, as these patients have poor outcomes compared to HPV-positive HNSCC patients. The identification of potential biomarkers in this subset of HNSCC would enable clinicians to diagnose the disease at an earlier stage and predict prognosis and treatment response.

For this reason, a recent study by Tawk et al. [17] developed a reliable and robust methylome-based classifier to identify HPV-negative HNSCC patients at risk for loco-regional recurrence and all event progression after post-operative radiochemotherapy (PORT-C). The study identified a 38-methylation probe-based HPV-negative Independent Classifier of Disease Recurrence (HICR) with a high prognostic value for loco-regional recurrence, distant metastasis, and overall survival. Among 38-differentially methylated probes (DMPs), 26 had functional annotations, while 12 probes lacked functional annotations. From the annotated probes, two probes were mapped to the *PCDHB4* transcription start site. Two probes mapped to the *INPP5A* gene body.

Aberrant patterns of methylation for these genes have not been reported in other tumor types. However, inositol polyphosphate 5-phosphatase (*INPP5A*) has been shown to have tumor-suppressive effects. In a nude mice model, its silencing was shown to increase intracellular inositol polyphosphate 5-phosphatase (*INPP5A*) and inositol 1,3,4,5-tetrakisphosphate, leading to cell transformation and tumor formation [18]. Interestingly, the loss of *INPP5A* was shown to be associated with decreased OS in recurrent/metastatic cutaneous squamous cell carcinoma [19]. 

Among the 38-methylation probes, several associated genes, such as *DCC*, *EOMES*, *SIM1,* and *HOXC9,* have also been shown to be hypermethylated in head and neck, bladder, cervical, and breast cancer, respectively [20,21,22].

Multivariate analysis of the same study by Tawk et al. [17] also showed that HICR high-risk tumors were enriched for younger patients with hypoxic tumors (15-gene hypoxia signature) and elevated 5-miRNA scores. The 5-miRNA signature (hsa-let-7g-3p, hsa-miR-6508-5p, hsa-miR-210-5p, hsa-miR-4306, and hsa-miR-7161-3p) was previously shown as a strong and independent prognostic factor for disease recurrence and survival of patients with HPV-negative HNSCC [23]. In HICR high tumors, the 26-hypoxia gene signature is an independent predictor of LR and PD (progressive disease), and a high 5-miRNA score was seen in 56% of the HICR high tumors. In HICR low-risk tumors, higher levels of CD3 and PD-L1 markers by IHC were prognostic of better treatment outcomes. 

In conclusion, the study proposed a 38-probe DNA methylation signature to classify HPV-negative HNSCC with the potential to stratify patients for therapy intensification/de-escalation strategies. Prospective validation of HICR is currently ongoing (Clinicaltrials.gov identifier NCT02059668). The aim of this clinical trial is to validate the prognostic impact of HICR on loco-regional control of locally advanced head and neck cancer after definitive or adjuvant chemoradiation. The primary objective is to evaluate local–regional recurrence-free survival (RFS) after two years for patients with locally advanced head and neck cancer who received definitive or adjuvant chemoradiation. A secondary objective is the evaluation of disease-free survival and metastases-free and overall survival.

The studies reviewed in Section 2.1 and Section 2.2 bring out the potential role of DNA methylation profiles as biomarkers, specifically, biomarkers in tumorigenesis, and also as a tool for stratifying patients at risk for recurrence or disease progression. These tools can help diagnose the disease early and in order to prepare treatments in a more effective way depending on the patient’s risk score. In the next sections, I will focus on the role of DNA methylation profiles as potential tumorigenesis, diagnostic, and prognostic biomarkers in both HPV-positive and HPV-negative HNSCC, including the subgroups of HNSCC and OPSCC (Table 1).

## 3. DNA Methylation as Potential Biomarkers

### 3.1. Differentially Methylated Genes That May Be Involved in Tumorigenesis of HPV-Positive HNSCC

While p16-IHC is a gold-standard test to diagnose HPV-positive HNCC, there are times that results can be equivocal and require HPV DNA detection with a PCR or by in situ hybridization (ISH) as an adjunct method to p16-IHC. Additionally, discordance between p16-IHC and HPV DNA detection is always present, which makes it harder for clinicians to assess the staging of the disease and the prognosis. Therefore, additional biomarkers that can detect the disease early would be helpful in a case like this.

A study by Hinic et al. [14] identified potential players that might contribute to the tumorigenesis of HPV-positive HNSCC. This includes the previously mentioned top two hypomethylated and upregulated genes in HPV-positive HNSCC: *SYCP2* (synaptponemal complex protein 2) and *TAF7L* (TATA-box binding protein associated factor 7-like). *SYCP2* is a testis-specific human gene and is associated with impaired meiosis [24]. It has been shown that dysregulation of *SYCP2* predicts the early stage of HPV-positive OPSCC. A study by Masterson et al. [15] used whole transcriptome analysis of locally advanced primary OPSCC in a prospective clinical trial. *SYCP2* showed the most consistent fold change by 3.1 from the baseline observed in premalignant tissues. The study suggested aberrant expression of this protein may contribute to genetic instability during HPV-associated cancer development [15]. Additional studies that can compare the methylation status of *SYCP2* in normal tissues vs. premalignant tissues will answer the question of whether the hypomethylation status of *SYCP2* itself has the potential as a biomarker in the tumorigenesis of HPV-positive HNSCC. *TAF7L* is an X-linked gene that is normally expressed in testis and spermatogonia [25]. *TAF7L* plays a role in the regulation of the transcription factor IID (TFIID) during male germ cell differentiation, and it has also been implicated in activator protein transcription regulation as a possible cofactor that binds to c-Jun [26,27]. *TAF7L* has been shown to be upregulated and involved in breast cancer development [16]. Further transcriptome analysis of *TAF7L* in OPSCC compared to normal tissues is needed to explore the potential function of *TAF7L* in the tumorigenesis of OPSCC. While more studies need to be conducted to investigate the further role of these genes in HNSCC, screening for circulating tumor DNA from peripheral blood using a panel of these genes may serve as an additional screening tool other than p16-IHC or HPV DNA detection for HPV-positive HNSCC.

### 3.2. DNA Methylation as a Potential Diagnostic Biomarker

While an HPV test is available for diagnosing HPV-positive HNSCC, the majority of the newly diagnosed patients are found with locally advanced and aggressive disease. Because of this, other measurable tools that could detect the disease early would be extremely helpful in managing the disease as early as possible, especially for HPV-negative HNSCC patients who have poor outcomes compared to HPV-positive HNSCC patients. In the next section, key genes that have been reported as potential diagnostic biomarkers are described.

#### 3.2.1. *EDNRB* and *DCC*

The *EDNRB* gene is located at 13q22.3 and encodes the B-type endothelin receptor (G protein-coupled receptor) that activates a phosphatidylinositol–calcium second messenger system. The *DCC* (deleted in colorectal cancer) gene is located at 18q21 and encodes a transmembrane protein with structural similarity to neural cell adhesion molecule (NCAM), which is involved in the differentiation of epithelial and neuronal cells [28,29]. A previous study by Demokan et al. [30] showed the promoter hypermethylation of *EDNRB* was present in 67.6% of salivary rinses from HNSCC and was suggested as a potential biomarker for diagnosing HNSCC [30]. Later, a prospective study by Schussel et al. [28] evaluated the hypermethylation status of *EDNRB* and eight additional genes (*DAPK*, *DCC*, *MINT-31*, *p16*, *MGMT*, *CCNA1* and *PGP 9.5*) in the salivary rinses of 191 patients. Among these genes, *EDNRD* and *DCC* hypermethylation identified dysplastic/cancer with a sensitivity of 46% and specificity of 72%, which were similar to the sensitivity of 56% and specificity of 66% in the case of identification by expert clinicians based on lesion examination. While a biopsy by expert clinicians is a gold-standard method for diagnosing the disease, a low-invasive approach like salivary rinses can be useful since it can be obtained easily by non-expert clinicians. They can screen populations who are at high risk for developing HNSCC or detect recurrence for those with a history of prior HNSCC [28]. However, a larger prospective cohort study is necessary to further confirm the potential role as a diagnostic biomarker.

#### 3.2.2. Differentially Methylated DNA Regions (DMRs) in HPV-Positive OPSCC

To investigate other measurable DNA alterations as potential diagnostic biomarkers other than HPV status in HPV-positive HNSCC, a genome-wide quantitative DNA methylation profiling was investigated [31]. The study used two independent cohorts, including 50 HPV-positive OPSCC patient samples and 25 normal samples. Normal tissue samples of the oropharynx were obtained from uvulopharynoplasty surgical samples in cancer-affected controls. The study discovered 51 Differentially Methylated Regions (DMRs) in the HPV-positive OPSCC patient samples that had significantly higher methylation levels compared to normal samples. A total of 51 DMR candidates were validated in the TCGA cohort that contained 48 HPV-positive OPSCC and 6 normal samples. Out of the 50 analyzed DMRs, 49 DMRs were significantly hypermethylated in HPV-positive OPSCC. In addition, the clustering of 51 DMRs showed significantly higher DNA methylation levels in HPV-positive OPSCC compared to the normal samples and non-HPV-related HNSCC. The study further validated the top 20 DMRs in another independent cohort of 24 HPV-positive OPSCC samples and 22 normal samples. These DMRs include the following genes: *KCNA3*, *EMBP1*, 362 *CCDC181*, *DPP4*, *ITGA4*, *BEND4*, *CTNND2*, *ELMO1*, *SFMBT2*, *C1QL3*, *MIR129-2*, 363 *ATP5EP2*, *OR6S1*, *NID2*, *HOXB4*, *ZNF439*, *ZNF93*, *VSTM2B*, *ZNF137P*, and *ZNF773.* In conclusion, the study suggests that the top 20 DMRs might be potentially helpful to use as an additional diagnostic test in OPSCC when the result of p16-IHC is equivocal or if there is discordance between p16-IHC and HPV DNA tests. It is important to investigate the potential role of a diagnostic biomarker in a prospective clinical trial that can also incorporate treatment to explore if these DMRs have prognostic implications as well.

#### 3.2.3. *Prominin 1* (*PROM1*)

The *PROM1* gene is located at 4p15.32 and encodes a pentaspan membrane glycoprotein, which is expressed in neuronal and hematopoietic stem cells. PROM1, also known as CD133, is currently considered a valuable marker for stem cells and cancer stem cells (CSCs) [32]. The study by Hu et al. [33] investigated the association between *PROM1* promoter methylation and HNSCC. The study used methylation profiles from 528 HNSCC and 50 normal tissues in the TCGA data portal [33]. Among the HNSCC tissues, 115 tissues have known a HPV status (74 HPV negative and 41 HPV positive). The analysis of methylation profiles showed that the hypermethylation of *PROM1* was significantly higher in the HNSCC samples compared to the normal tissues. The study also showed that *PROM1* promoter methylation levels were associated significantly with age, smoking history, and T stage, which suggests a potential role in the invasion/progression of the disease. No significant correlations with other clinicopathological characteristics, such as alcohol history, tumor site, HPV status, or nodal metastasis, were present. Additionally, the hypermethylation of *PROM1* was associated with poor RFS and OS, but this association was not significantly different between HPV-positive and HPV-negative HNSCC patients. Overall, the study suggests that the methylation status of *PROM1* may possess a dual function as both a diagnostic and prognostic biomarker. Since *PROM1* promoter hypermethylation seems to be a general methylation signature in HNSCC regardless of anatomical location and HPV status, prospective clinical trials with large sample sizes and correlative studies analyzing *PROM1* promoter methylation status are warranted to further explore the potential role of dual biomarkers in any subgroups of HNSCC.

### 3.3. DNA Methylation as a Potential Prognostic Biomarker

In the next section, this study summarizes evidence that aberrantly methylated DNA profiles in HNSCC may have potential as a prognostic biomarker. A prognostic biomarker can give us information on the possibility of relapse or disease progression, which will help detect relapse or disease progression at an early stage. An extensive list of methylated genes and proteins with prognostic potential are present in different cancer types, but this study focuses on key methylated genes that have been valued as a potential prognostic biomarker in both HPV-positive- and HPV-negative HNSCC, including subgroups of HNSCC and OPSCC.

#### 3.3.1. Aberrantly Methylated Genes as Potential Prognostic Biomarkers in OPSCC

##### *p16^INK4A^* and *p14^ARF^* (Cell Cycle Regulation Genes) 

The *CDKN2A* gene encodes two tumor suppressor genes, *p16^INK4A^* and *p14^ARF^*. Aberrant promoter hypermethylation of these genes has been observed in oropharyngeal cancer as well as premalignant oral lesions [34,35]. Promoter methylation of *p16^INK4A^* was shown to be associated with increased disease recurrence and acts as an independent predictor for a worse prognosis [34]. However, *p14^ARF^* methylation is associated with a lower recurrence rate in oral cancer patients with a good clinical outcome [34]. More recently, a systematic meta-analysis comparison amongst HNSCC, premalignant lesions, and normal controls showed a significant increase in the frequency of *CDKN2A* methylation during HNSCC tumorigenesis. In addition, *CDKN2A* hypermethylation was significantly associated with shorter overall survival (OS) and recurrence-free survival (RFS), which suggests CDNK2A could be a potential prognostic biomarker. However, the finding was not stratified by HPV status [35]. Since a novel prognostic biomarker is urgently needed in HPV-negative HNSCC specifically, a prospective study to further investigate the role of *CDKN2A* hypermethylation in HPV-negative HNSCC patients as a potential prognostic biomarker would be imperative to advance the management of HPV-negative HNSCC patients.

#### 3.3.2. Aberrantly Methylated Genes as Potential Prognostic Biomarkers in HPV-Negative HNSCC

##### *MGMT* 

The *MGMT* (methylguanine methyltransferase) gene is located at 10q26 and encodes for a protein that repairs DNA. *MGMT* promoter methylation has been known as a predictive biomarker for the response to radiotherapy and a prognostic biomarker for adjuvant alkylating chemotherapy in patients with glioblastomas [36]. Along with hypermethylation in promoter regions of tumor suppressor genes, *MGMT* has been observed in various types of cancers including oral cancer [37,38,39,40,41]. A study by Reis et al. [42] investigated whether there is an association between the hypermethylation of *MGMT* and prognosis (OS: overall survival, DFS: disease-free survival) in 72 HPV-negative OPSCC patients. Using a methylation-specific PCR, promoter methylation was evaluated, and the hypermethylation frequencies of *MGMT* in the 72 tumors was 19.44%. The study also revealed that patients with hypermethylation of *MGMT* had a better 2-year OS compared to patients without hypermethylation. Since the *MGMT* promoter is known as a prognostic biomarker for alkylating chemotherapy in glioblastoma patients, the study also investigated an association between the use of cisplatin and *MGMT* promoter methylation; however, there was no significant association. This could be due to the small sample size and different anatomical sites included in the study. A prospective clinical trial with a larger sample size including subjects whose tumors are from various anatomical sites is warranted to confirm the prognostic value of *MGMT* promoter hypermethylation in HPV-negative OPSCC.

##### *COL1A2* 

The *COL1A2* gene is located at 17q21.33 and encodes pro-alpha1 chains of type I collagen, which is found in most connective tissues. Hypermethylation of *COL1A2* has been observed in different types of cancers, including breast, melanoma, and medulloblastomas [43,44,45]. A study by Misawa et al. [46] was carried out to examine the methylation status of 30 tumor-related genes (TRGs) to assess recurrence and patient survival in HPV-negative HNSCC. A total of 178 HPV-negative head and neck cancers that originated in the hypopharynx, larynx, and oral cavity were included in the study. The results of the study showed that methylation of *COL1A2* was positively correlated with recurrence in hypopharyngeal cancers. Methylation of *COL1A2* was also shown to be correlated with poor survival for both hypopharyngeal and laryngeal cancer. These associations were not observed in oral cavity cancer. The study suggests that *COL1A2* methylation status may serve as an important site-specific biomarker for the prediction of clinical outcomes in patients with HPV-negative HNSCC.

**Table 1 cancers-15-04685-t001:** Genes that are aberrantly methylated in HNSCC and their roles as potential biomarkers.

HPV Status	Methylation Status/Gene Name	Biological Pathway/Function	Potential Biomarker (Tumorigenesis, Diagnostic, Prognostic)
**HPV+**	**Hypermethylated**		
	*CCNA1*, *RASSF1*, *CDKN2A*	Cell cycle regulation and apoptosis	
	*CDH* (*1*, *8*, *11*, *13*, *15*, *18*, *19*, *23*)	Cellular adhesion	
	*CADM1*, *ITGA4*	Cellular adhesion	
	*TIMP3*, *ELMO1*, *CTNNA2*	Cellular migration	
	*RXRG*, *GATA4*	Differentiation	
	*SULF1* (*Sulfatase 1*)	Growth factor signaling, tumorigenesis	
	*GPCR* (*HCRTR2*, *GALR1*, *HCRTR1*, *TACR1*, *NTSR1*)	Involved in cAMP and phosphatidylinositol signaling pathway	
	20 genes as methylation markers (listed in Section 3.2.2)		**Diagnostic:** 20 DMRs, higher DNA methylation compared to normal tissues, and HPV-negative HNSCC [31]
	**Hypomethylated**		
	*SYCP2*	Associated with impaired meiosis	**Tumorigenesis:** Significant increase of gene expression in premalignant and OPSCC compared to normal tissues [15]
	*TAF7L*	Regulate the binding of the TFIID/RNA polymerase II (RNAP II) complex	**Tumorigenesis:** Involved in tumorigenesis of breast cancer [16]; no report on HNSCC yet
**HPV−**	38-differentially methylated probes (DMPs)		**Prognostic:** High prognostic value for local regional recurrence, distant metastasis, and OS for locally advanced HNSCC [17]
	*MGMT*	Tumor suppressor	**Prognostic:** Hypermethylation of MGMT had a better 2-year OS [42]
	*COL1A2*	Encodes fibrillary collagen	**Prognostic:** Hypermethylation of COL1A2 is associated with disease-free survival specifically in hypopharyngeal and laryngeal cancer [46]
**Unspecified**	**Hypermethylated**		
	*EDNRB*	EDNRB: Encodes B-type endothelin receptor	**Diagnostic:** Hypermethylated EDNRB and DCC in precancerous lesions and oral cavity cancer [28,29]
	*DCC*	DCC: Tumor suppressor	
	*Prominin 1* (*PROM1*)		**Diagnostic:** Hypermethylated PROM1 compared to normal tissue [32] **Prognostic:** PROM1 promoter methylation is associated with poor RFS and OS [32]
	*CDKN2A*	Encodes p16^INK4A^ and p14^ARF^; both function as tumor suppressor	**Prognostic:** Promoter hypermethylation is associated with shorter OS and RFS [35]
	*CDKN2A* (*p16^INK^* gene-specific)	Activates RB-dependent cell cycle arrest	**Prognostic:** Promoter hypermethylation is associated with increased disease recurrence in oral cavity cancer [34]
	*CDKN2A* (*p14^ARF^* gene-specific)	Activates the tumor suppressor gene p53 by inhibiting MDM2	**Prognostic:** Promoter hypermethylation is associated with decreased disease recurrence in oral cavity cancer [34]

## 4. Role of DNA Methylation Profiles in Relation to Response in Immunotherapy

Recent studies have demonstrated that certain patterns of DNA methylation could be used as a predictive biomarker in response to PD-1 (programmed-death-1) inhibitors and CTLA-4 (cytotoxic T-lymphocyte-associated protein 4) inhibitors in different cancer types. For example, a study by Starzer et al. [47] analyzed tumor DNA methylation profiles in relation to immunological parameters and the response to PD-1 inhibitors in patients with sarcomas. The most differentially methylated genes between responders and non-responders were found to be involved in pathways such as Rap1 signaling and pathways in cancer, focal adhesion, adherens junctions, and extracellular matrix (ECM)–receptor interaction. These different DNA methylation profiles should be further investigated as a potential predictive biomarker for the response to PD-1 inhibitors in sarcomas and also in other types of cancer.

Another study by Fietz et al. [48] investigated the association between *CTLA4* methylation status and anti-CTLA4 immunotherapy (ipilimumab) response. The result of this study showed that lower methylation levels were observed in responders compared to non-responders. In addition, lower CTLA4 methylation levels were associated with increased progression-free survival (PFS). Interestingly, CTLA-4 protein expression was not associated with the treatment response. The study suggests that *CTLA4* methylation could serve as a predictive biomarker for the response to anti-CTLA-4 immunotherapy in melanoma [48].

While limited studies have been conducted to investigate the role of DNA methylation as a predictive biomarker in HNSCC, several recent studies suggested DNA methylation as a potential predictive biomarker in HNSCC particularly in immunotherapy. The tumor necrosis factor receptor superfamily members 4 (*TNFRSF4*, *OX40*) and 18 (*TNFRSF18*, *GITR*, *AITR*) have been under investigation as potential targets for the immunotherapy of various cancers, including HNSCC [49,50]. A study by Loick et al. [49] investigated a detailed description of the DNA methylation landscape within *GITR* and *OX40* (both located in close proximity on chromosome *1 p36.33* in HNSCC) [49]. The study conducted a broad correlation analysis of the DNA methylation of 46 CpG sites within the *GITR/OX40* gene locus in HNSCC and normal adjacent tissues provided by TCGA (the Cancer Genome Atlas). The study found that there were significant methylation differences between normal adjacent and tumor tissues and also between HPV-positive- and HPV-negative HNSCC. At 17 CpG sites, the DNA methylation levels in HPV-negative tumor tissue exceeded the methylation levels from HPV-positive tumors, particularly in the CpG sites located in the promoter region of *GITR*. The study found that there were significant correlations between specific CpG sites and signatures of immune cell infiltrates and interferon-γ depending on the genomic positions of the analyzed CpG sites. A positive correlation of mRNA expression of *GITR* and *OX40* and DNA methylation with OS was also observed. However, these correlations need to be further investigated in HPV-positive- and HPV-negative HNSCC, separately. In conclusion, the study provides a framework to prospectively test specific CpG sites as potential prognostic biomarkers. In addition, CpG site-specific *GITR* and *OX40* methylation might be used to identify HNSCC patients who could benefit from immunotherapy. 

Another recent study by Starzer et al. [51] investigated the DNA methylation profiles of tumors from patients with R/M HNSCC treated with PD-1 inhibitor. In the study, 37 patients with R/M HNSCC were included. The median number of prior systemic therapies was 1 (range 1–4). A total of 5 out of 37 (13.5%) patients achieved an objective response to PD-1 inhibitor. Median progression-free survival and median overall survival times were 3.7 months and 9.0 months, respectively. Microarray analyses showed the methylation signature including both hypomethylation and hypermethylation was predictive of the response to immunotherapy. Differentially methylated genes between responders and non-responders were associated with several molecular pathways, including axon guidance (e.g., *BMPR1B*, *CAMK2D*, *EPHNA6*, *NTNG1*), hippo signaling (*AFP*, *BMP7*, *GLI 1*), pathways in cancer (*IL2RA*, *MAPK10*, *IGF1R*, *AKT3*, *MLH1*, *COL4A1*), and MAP signaling (e.g., *TAOK3*, *STK3*, *IL1RAP*, *MAP3K1*). While there was no specific inflammation pathway that showed differential methylation, an inflammatory-related gene such as *IL2 receptor alpha* (*IL2RA*) included in the pathways in cancer was affected by the differential methylation between non-responders and responders. Previously, a preclinical study using tumor-targeting IL2 treatment showed a tumor-infiltrating CD8+ T cell response and effective tumor control [52]. Interestingly, none of these genes seemed to be overlapping with the other DNA methylation profiles reviewed here as potential biomarkers. This could be due to the low sample size, differences in the anatomical origin of the tissues used in the analysis, or it could be that the DNA methylation profiles in HNSCC patients changed with exposure to immunotherapy. 

Given the lack of reliable predictive biomarkers in HNSCC for immunotherapy, the results of the study deserve a validation study. Using patient samples from large phase 3 clinical trials of immunotherapy in HNSCC, the validation study should be carried out retrospectively. However, the problem with retrospective validation is that results can be confounded by a number of previous treatments, such as radiotherapy and systemic therapy, that can influence the tumor’s immune microenvironment. For this reason, analyzing data from previous treatment modalities would be necessary. Once validated, a prospective study will further guide the development of methylation signatures as a predictive biomarker for immunotherapy response.

Recent studies have revealed that APOBEC (apolipoprotein B mRNA editing enzyme, catalytic polypeptide-like), a single-stranded-DNA cytosine-to-uracil deaminase, plays an important role not only in carcinogenesis and tumor progression but also in immune activation in different types of cancer [53]. The study by Liu et al. [54] investigated the role of APOBEC in HNSCC using 530 patients and 74 normal people from the TCGA database. The study found that the APOBEC protein family member APOBEC3H was significantly upregulated in HNSCC patients compared to normal people, and a higher level of APOBEC3H was observed in HPV-positive compared to HPV-negative patients. In addition, patients with higher APOBEC3H levels had a better OS. Interestingly, tumors with high APOBEC3H levels revealed genome-wide DNA hypomethylation. Specifically, *CXCL10*, which plays an important role in CD8 T cell infiltration into the tumor microenvironment, was substantially hypomethylated. Both *CXCL10* and *APOBEC3H* expression levels were positively correlated with CD8+ T cell infiltration, which showed a better prognosis. The study strongly suggests APOBEC3H might regulate immune activity through demethylation of *CXCL10*, which results in infiltration of the CD8+ T cell and ultimately a better prognosis. Since higher APOBEC3H is observed in HPV-positive HNSCC, one could consider validating the results in a prospective clinical trial of immunotherapy in HPV-positive HNSCC patients. This could further guide the development of APOBEC3H and its mediated *CXCL10* methylation status as a potential predictive biomarker for response to immunotherapy. Differentially methylated genes in relation to the response in immunotherapy are outlined in Table 2.

## 5. Role of DNA Methylation in Modulating the Tumor Immune Microenvironment (TIME) in HNSCC

Various studies have suggested that DNA methylation is involved in genetic alterations in tumor cells and in the tumor microenvironment [55,56]. Epigenetic drugs like DNA methylatrasnferase-1 (DNMT1) inhibitors have been shown to have positive effects in cancer treatment, but more recent studies demonstrated that DNMT inhibitors also have the potential to improve the effect of immune checkpoint blockade therapy by modulating the TIME [9]. However, the role of DNMT1 in the TIME in HNSCC is not well understood. The study by Yang et al. [57] investigated the expression levels of DNMT1 and its association with prognosis by analyzing human oral squamous cell carcinoma (OSCC) tissue microarrays. The authors developed two different types of immunocompetent mouse OSCC models to explore the effects of DNMT1 inhibitors in the TIME. 

The study showed that DNMT1 inhibition reduced tumor size in immunocompetent mouse OSCC models. The results showed DNMT inhibition increased the tumor infiltration of CD3, CD4, and CD8 T cells as well as decreased the expression levels of immunosuppressive factors, such as *PAK2* and *VISTA,* compared to the control group. These findings suggest that DNMT1 inhibition can be a potential novel therapy in combination with immunotherapy to enhance the immune response by modulating the TIME.

Multiple clinical trials using antibodies targeting CTLA-4, such as ipilimumab and tremelimumab, are currently under investigation for treatment of HNSCC. CTLA-4 belongs to the B7-CD28 superfamily, which is known to regulate immune response via ‘costimulatory’ or ‘coinhibitory’ signals [58]. A better understanding of the epigenetic regulation of the B7-CD28 superfamily members, such as B7, CD80, and CD86 (B7 ligands), CTLA-4, and ICOS, will help to identify biomarkers for response prediction to anti-CTLA-4 immunotherapy. 

For this reason, a study by de Vos et al. [59] investigated the role of DNA methylation of the encoding genes *CD28*, *CTLA4*, *ICOS*, *CD80,* and *CD86* in modulating the TIME of HNSCC. The study investigated the methylation of these genes at single CpG resolution (51 CpG sites) in a cohort of HNSCC and normal adjacent tissue samples provided by the Cancer Genome Research Atlas. The study results showed that *CTLA-4*, *CD28*, *CD80*, *CD86,* and *ICOS* expression levels are epigenetically regulated by DNA methylation. Depending on the specific CpG site, methylation of these genes was correlated with CD8+ T cell infiltration, tumor mutational burden, and IFN-γ signature. The result suggests that DNA methylation directly or indirectly modulates the TIME and provides evidence to test DNA methylation as a potential biomarker for the prediction of the response to CTLA-4 immune checkpoint inhibitors.

Another study demonstrated the role of squalene cyclooxygensase (SQLE) in the TIME of HNSCC [60]. The results of this study showed that high SQLE expression is found to promote cell proliferation of HNSCC and associates with the T stage in HNSCC. The methylation levels of three CpG sites in S_Shore were significantly demethylated in HNSCC compared to normal samples. Copy number amplification and DNA demethylation synergistically promote the overexpression of SQLE expression, and this was associated with a poor prognosis. In addition, mRNA expression/copy number alterations were negatively correlated with an infiltration of CD8+ T cells, follicular helper T cells, regulatory T cell infiltration, and mast cell activation and positively correlated with the infiltration of M0 macrophages and resting mast cells in HNSCC. Overall, the result suggests that SQLE could be further investigated as a potential predictive biomarker and a potential pharmaceutical target to improve the immune response in the treatment of HNSCC. Differentially methylated genes in relation to TIME are outlined in Table 2.

**Table 2 cancers-15-04685-t002:** Role of DNA methylation in modulating tumor immune microenvironment (TIME) and as potential biomarker in response to immunotherapy.

Methylation Status of Specific Gene/Biological Function of Gene or Involved Pathways	Modulation in TIME	Potential Biomarker in Response to Immunotherapy
*GITR* promoter CpG-specific methylation/tumor necrosis factor receptor superfamily member 4	Positive or negative correlation with T cell infiltration; interferon-γ depends on CpG sites [49]	Specific CpG sites of *GITR* as predictive biomarker [49]
*OX40* promoter CpG-specific methylation/tumor necrosis factor receptor superfamily member 4	Positive or negative correlation with T cell infiltration; interferon-γ depends on CpG sites [49]	Specific CpG sites of *OX40* as predictive biomarkers [49]
Differently methylated genes (hyper- and hypomethylated)/axon guidance, hippo signaling, pathways in cancer, and MAP signaling		DNA methylation profiles as a predictive biomarker in response to PD-1 inhibitors [51]
*APOBEC3H*-mediated demethylation of *CXCL10*/ Chemokine	Increase CD8+ T cell tumor infiltration [54]	Higher APOBEC3H protein level is associated with better OS [54]
*SQLE* CpG-specific demethylation	Negative correlation with T cell infiltrates [60]	

## 6. Ongoing Clinical Investigations in DNA Methylation Profiles and Demethylation Therapy in HNSCC Patients

In this section, currently ongoing clinical trials investigating DNA methylation profiles in HNSCC and also demethylation therapy as a novel therapy to treat HNSCC patients are summarized (Table 3). There is one case-control prospective study at Aalborg University Hospital in Denmark to investigate specific methylation profiles in head and neck cancer (Clinicaltrials.gov identifier NCT04567056). Peripheral blood samples were collected for methylation analysis and compared between verified HNSCC patients and a control group (without active or earlier cancer). The study opened in 2020, is currently active, and aims to enroll about 300 patients. 

So far, DNMT inhibitors such as azacitidine and decitabine have been approved for the treatment of myelodysplastic syndrome (MDS) and acute myeloid leukemia (AML) [61,62]. In this section, ongoing clinical trials using azacitidine or decitabine as a monotherapy or with chemotherapy/immunotherapy in HNSCC are reviewed (Table 2). 

A phase 1 window of opportunity clinical trial (Clinicaltrials.gov identifier NCT02178072) investigated the activity of 5-azacitidine in patients with HPV-positive HNSCC. This study was pursued based on the result of a mouse model of HPV-positive HNSCC which showed reduced expression of HPV genes, stabilization of p53, and induced p53-dependent apoptosis in HNSCC [63]. Eligible patients were patients with newly diagnosed, surgically resectable HNSCC. Azacitidine was administered intravenously at 75 mg/m^2^/d for 5 or 7 days. The primary objective was to determine the proportion of HPV-positive patients in whom 5-azacitidine increases APOBEC RNA expression. The secondary objective was to investigate the safety and clinical activity of 5-azacitidine. The study was completed, but the results are not published yet.

There is an ongoing clinical study to investigate if 5-azacytidine makes HPV-associated head and neck cancer more sensitive to treatment with nivolumab. This is a phase 0 study, which is a three-arm window trial randomizing patients to pre-operative treatment with 5-azacytidine alone, to nivolumab alone, or to a combination of 5-azacytidine and nivolumab (Clinicaltrials.gov identifier NCT05317000). The primary endpoint is an immune-related pathologic response according to the criteria of Cottrell et al. [64]. The secondary endpoint is augmentation of tumor infiltration of the tumor microenvironment as determined by a quantitative immunofluorescence score (QIF) measuring CD3+ lymphocytes and granzyme B expression.

Patients were eligible with T1-3, N0-2, M0 p16-positive squamous cell carcinoma of the oropharynx deemed resectable by surgery. Patients in Arm A received 5-azacytidine 75 mg/m^2^ IV once daily for days 1–5. Patients in Arm B received nivolumab 240 mg IV on days 1 and 15. Patients in Arm C received 5-azacytidine as described above and received nivolumab 240 mg IV on days 2 and 16. In arms A and B, surgery was performed during the period of days 16 to 18 and in Arm C during the period of days 17 to 18. The study enrolled 8 patients in 5-azacytidine monotherapy and 20 patients per arm in the nivolumab or 5-azacytidine/nivolumab combination groups. The study started in March of 2023.

Another DNMT inhibitor, decitabine, is being investigated in clinical trials (Clinicaltrials.gov identifier NCT03019003). This is a phase 1b/2 study to assess the safety and efficacy of oral decitabine (ASTX727) and durvalumab (MEDI4736) in combination in R/M HNSCC. Patients are eligible with histologically confirmed R/M HNSCC (oral cavity, oropharynx, hypopharynx, or larynx) that has progressed during or after treatment with anti-PD-1, anti-PD-L1, or anti-CTLA4 monotherapy. The study aims to define the highest effective dose of decitabine in combination with durvalumab (MEDI4736) to improve the immune system in order to recognize and target head and neck cancer cells. Decitabine is also being investigated in the treatment of nasopharyngeal cancer (Clinicaltrials.gov identifier NCT03701451). The study aims to investigate the efficacy and toxicity of decitabine and cisplatin as induction chemotherapy for three cycles followed by concurrent chemoradiotherapy in the treatment of regionally advanced nasopharyngeal carcinoma. The study is currently active; no preliminary results are available.

## 7. Conclusions

This review summarizes the current knowledge about DNA methylation profiles in HNSCC. It strongly suggests the role of differential DNA methylation profiles as potential diagnostic, prognostic, and predictive biomarkers. While different DNA methylation profiles have been reviewed here as potential biomarkers in HNSCC, it is imperative to think about which specific DNA methylation profile might be more clinically valuable than others. For example, differently, methylated genes such as *SYCP2* and *TAF7L* are suggested as potential biomarkers in the tumorigenesis of HPV-positive HNSCC [15,16]. These biomarkers can be valuable in clinical settings to detect disease in its early stages either as an initial diagnosis or as recurrent disease, especially when the tumor has either an equivocal p16-IHC result or discordance between p16-IHC and HPV DNA tests result. Regarding the role of DNA methylation profiles in HPV-negative HNSCC, hypermethylation of *MGMT* and *COL1A2* is another example that can be considered a clinically valuable prognostic factor since there is a lack of biomarkers for HPV-negative HNSCC in general. Additionally, the future application of 38 differentially methylated probes as biomarkers to classify disease recurrence, distant metastasis, and overall survival [17] is also clinically valuable, as it may help stratify HPV-negative HNSCC patients. This could provide more personalized treatment based on methylation profiles with hopefully better clinical outcomes. 

In the current immunotherapy era, a novel biomarker to predict response and a novel therapy to improve immunotherapy response are urgently needed. This review highlights differential methylations of specific CpG regions of *GITR/OX40* as a potential predictive immunotherapy biomarker [49]. While ongoing clinical trials are actively investigating demethylation therapy as a potential novel therapy to improve immunotherapy response, a major hurdle in implementing demethylation therapy in head and neck clinical trials is due to the lack of biomarkers for response to treatment either in locally advanced or R/M HNSCC. With more understanding of the role of DNA methylation profiles as potential biomarkers, I anticipate better-designed clinical trials will ultimately improve treatment responses in HNSCC patients.

## Figures and Tables

**Table 3 cancers-15-04685-t003:** Overview of HNSCC clinical trials using DNMT inhibitors (monotherapy or combination).

Reference/NCT#	Status	Phase	DNMT Inhibitor	Chemotherapy/ Immunotherapy	Study Duration	Disease Target	Result
NCT02178072	Recruiting	Window study	Azacitidine		2018– completed	HNSCC (HPV+ resectable disease)	**Pending**
NCT05317000	Recruiting	Window study	Azacitidine	Nivolumab	2023– ongoing	HNSCC (HPV+/− resectable disease)	**Pending**
NCT03019003	Recruiting	1b	Decitabine	Durvalumab	2017– ongoing	HNSCC (R/M, refractory to immune checkpoint inhibitor)	**Pending**
NCT03701451	Recruiting	1b/2	Decitabine	Cisplatin	2018– ongoing	Locally advanced nasopharyngeal carcinoma (NPC)	**Pending**

Black, bold font: clinical trial ongoing, and results not available yet.

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
