# Peer review of "Role of DNA Methylation Profiles as Potential Biomarkers and Novel Therapeutic Targets in Head and Neck Cancer"

_cancers, 2023, doi:10.3390/cancers15194685_

Round 1
Reviewer 1 Report
The review entitled: Overview of DNA Methylation Profiles, Role as Potential Biomarkers and Novel Therapeutic Targets in Head and Neck Cancer
Type of manuscript: Systematic Review
Journal: Cancers
The manuscript highlights the value of DNA methylation profiles as biomarkers for diagnosis and prognosis in HNSCC. This article is of scientific value, however, there are some major concern which are missing and should be addressed. Below are my comments:
There is only one author of the work, in the simple summary and in the introduction section the phrases are used – “we”, please correct.
The topic does not fully correspond to the purpose of the work.
Gene names should be in italics, there is a mix in the article, please correct.
The article should contain more tables, figures to make the work more readable (the most important issues should be on figures and in the tables). I would recommend providing also a table summarizing the advantages and disadvantages of using DNA Methylation Profiles in HNSCC.
Organize HPV positive, negative, HNCSS, OPSCC, OSCC information.
In introduction, the clinical significance must be discussed as background information, and more comprehensive explanatory as results of various studies should be presented - the characterization, application, future perspectives.
What is the main role of DNA Methylation Profiles in HNSCC - specific and important points are missing in discussion. The article contains too long descriptions, not always important, not always corresponding to the purpose of the work - it should be shortened to the most important issues. Organize the information, also explain what may affect the differences in the results obtained (methods used, number of samples, type of control used, etc.)
Therefore, how effective are Methylation Profiles in HNSCC as biomarkers? Discuss based on evidences collected studies.
The conclusions should be improved, in many places it is a repetition of the presented studies. The conclusion is too long, many things are described as results of other studies of DNA Methylation Profiles as biomarker in HNSCC. It is not concusion of the manuscript – systematic review.
5.3.2.1.MGMT - in text the author describe not only information about MGMT gene but also about another genes, why?
TIME- explain abbreviation 5.2.2. – italic?Author Response
Thank you very much for you review. Please see attachment

Reviewer 2 Report
In the manuscript entitled “Overview of DNA Methylation Profiles, Role as Potential Biomarkers and Novel Therapeutic Targets in Head and Neck Cancer” the author presents a Systematic Review on the advances on HNSCC specific DNA methylation profiles and their as therapeutic targets.
The paper is well written, and that makes it very easy reading. It is quite a long review, with lots of information and as such a good starting point for those who want to learn more about DNA methylation in HNSCC.
Having said this, my main criticism is that this paper does not appear to fulfill the criteria for a systematic review. When (briefly) reviewing the PRISMA criteria I am missing important information. How exactly was the literature screening, and subsequent filtering carried out, and did the author focus on a specific time window? I would like to see (maybe as supplementary data) numbers: In how many papers did the search result. The search criteria were not completely clear to me and/or gave surprising results when I did a search myself. For some searches I got 0 results. On the other hand, I have to say that a large part of this manuscript (the first sections), do not really lend themselves for a systematic review.
Another issue was the confusing content of the manuscript in relation to its objectives as stated in the title. My confusion became clear in the section “conclusions”. In this section two different objectives of this review are discussed (L638-640 and L670-675). Although they are both about DNA methylation, there is no further connection. No overlap in genes that were discussed (as one would expect). Now I that the impression that I was reading two different reviews merged into one. The easiest way to solve this issue would be to delete section 3, or at least all the subsections. The most interesting part of the review is where you actually discuss the possible contribution of DNA methylation as biomarkers other than the already known distinction HPV+ or HPV-
Following additional small and more important comments in the order of appearance in the paper.
L61: is > are (> sign indicates “should be”)
L72: the sentence “Resent”: this principle is already known for several decades. When such a general statement is made the original discovery deserves to be mentioned. I think the references mentioned may not even be the first for HNSCC.
L77: ref 8 is a review on itself and does not provide any “evidence”
L78-86: I have some problems with this paragraph. The difference between HPV -positive and HPV-negative tumors is made rather easily. HPV-positive tumors have a relative good prognosis, in contrast to the HPV negative ones. Why should then a review focus on differences in DNA methylation profiles between these two groups. We already know how to distinguish these groups.
L99: specify which DNMT.
L102-112: As mentioned above, more information is needed to justify this review as a systematic review.
L113 Section 3 maybe informative and well written, but I fail to see how it is important, as we already have a marker for HPV-positive tumors. L206-207 is a good example of my problem. How will this knowledge help us in diagnostics, and thus how could this play a role as a biomarker on top of our screening for HPV positivity. In fact it seems the author agrees with me (L276)
L116 DNP > DNA (?)
L117: oncoproteins misspelled.
L124 correct this sentence (remove “that”?)
L139: section 3.1 is most of all a summary of differential methylated genes in HPV-positive HNSCC. In fact I personally do not think that such a section is suitable for a systematic review. As for many of these genes their methylation status in cancer is already known for a long time, this seems to be outside the scope of a systematic review. The methylation status of RASSF1 has already been describes early this century (Hasegawa_M et al).
L187: here the entity “oropharyngeal squamous cell carcinoma” is introduced. Later several paragraphs appear to focus on OPSCC. Why is this not mentioned in the introduction as a specific subgroup of HNSCC.
One could say that the interesting part of this review starts at line 280. Maybe the previous part could be shortened considerably.
L282: sentence needs correction??
L326-347. The question remains why we need additional markers to identify HPV-positive tumors.
L326: contribute to tumorigenesis.
L332: explain OPC
L351: “early is in great”. Please correct
L360: the questions comes to mind if 40% is high enough to be used in a diagnostic test. This could be discussed.
L370: how were these normal samples defined/obtained?
L382: What would we gain if this would turn out to be a good marker to identify HPV-positive HNSCC?
L384-396: In this study, is the HPV status unknown?
L394: A question that comes to mind: what criteria should a biomarker fulfill in order to be a clinical interesting biomarker. That would be an interesting addition to this review.
L436: “could be served as a prognostic in HPV” > “could serve as a prognostic marker in HPV”
L464: adnesion > adhesion
L488-495: It is unclear if this correlation is independent of HPV or not.
L497: another broken sentence
L497-507: A little more info would have been nice: Which genes were differential methylated, and have these genes been mentioned in other studies as well? Is there an overlap with other studies discussed in this review?
L638-640: This is a little different than what the title suggested. Now suddenly it is not about new biomarkers anymore.
L664: reference is missing.
L672: is > are
L670-675: a second objective of this review! The problem is that – in term of significant genes - I do not see any overlap between the first and the second goal. So in fact I am looking at two different reviews in one paper.
Given my major issues as stated in the beginning of this letter, I find it important to state at this point that I did exhaustively comment on the various small issues that I encountered.
In general terms, the paper is well written, and easy to read. However I noticed several spelling errors, seemingly missing words and /or grammatical incorrect sentences. These mistakes should be corrected as they make a sloppy impression.
Reviewer 3 Report
the paper consider how DNA methylation alters the gene expression, which drives tumorigenesis in cancer, and regulates tumor immune microenvironment.
The authours considered HPV positive and negative HNSCC and gene involved in cell cycle regulation and apoptosis; in cellular adhesion;in cellular migration;in differentiation and in G Protein-Coupled Receptor (GPCR) signaling.
They confirmed that HPV positive tumors have pathways involved in the extracellular matrix-interaction, cytokine production,cell cycle regulators, apoptosi.
there are a lot of ongoing clinical studies to investigate DNMT inhibitors
In this study,authours review current knowledge about DNA methylation in HPV positive HNSCC and HPV negative HNSCC ,the report is satisfactory
english is fine for me
Author Response
Thank you very much for your review.
Editing of English is completed. More information and interpretations are added to results.
Round 2
Reviewer 2 Report
The paper has been extensively modified, and it seems that the author has dealt with most of my criticisms. However, the large amount of "track-changes" has made it difficult to evaluate the document. Therefore I would like to ask the author to submit a clean version of the manuscript in addition to the present manuscript.
I have the impression that the English language will still need some attention.